# Preparation of Yellowing-Resistant Waterborne Polyurethane Modified with Disulfide Bonds

**DOI:** 10.3390/molecules29092099

**Published:** 2024-05-02

**Authors:** Guorong Li, Baorong Duan, Guorui Leng, Junjie Liu, Tong Zhang, Zhenwei Lu, Shanshan Wang, Jiale Qu

**Affiliations:** 1School of Pharmacy, Binzhou Medical University, Yantai 264003, China; liguorong004@163.com; 2Research Center for Leather and Protein, College of Chemistry & Chemical Engineering, Yantai University, Yantai 264005, China; duanbaorong@126.com (B.D.); m18678708615@163.com (T.Z.); wei2082762541@163.com (Z.L.); 3School of Special Education and Rehabilitation, Binzhou Medical University, Yantai 264003, China; rruiiiiii_@outlook.com; 4Department of Physics, Binzhou Medical University, Yantai 264003, China; liujunjie@bzmc.edu.cn; 5Key Laboratory of Coastal Environmental Processes and Ecological Remediation, Yantai Institute of Coastal Zone Research (YIC), Chinese Academy of Sciences (CAS), Yantai 264003, China

**Keywords:** disulfide, waterborne polyurethane, yellowing resistance, UV absorber UV–320, dynamic reversibility

## Abstract

Waterborne polyurethane, renowned for its lightweight properties, excellent insulation capabilities, and corrosion resistance, has found extensive application in fields such as construction, automotive, leather, and thermal insulation. Nevertheless, during operational usage, waterborne polyurethane materials, akin to other polymeric substances, are susceptible to oxidative aging manifestations like yellowing, cracking, and diminished mechanical performance, significantly curtailing their utility. Consequently, the synthesis of yellowing-resistant polyurethane assumes pivotal significance. This study integrates dynamic reversible reactions into the synthesis process of polyurethane by introducing the dynamic reversible compound 2-hydroxyethyl disulfide as a chain extender, alongside the incorporation of a UV absorber to enhance the polyurethane’s resistance to yellowing. When the disulfide bonds absorb heat, they undergo cleavage, yielding thiols that spontaneously recombine into disulfide bonds at ambient temperatures, allowing for the continuous breaking and reformation of disulfide bonds to absorb heat. Concurrently, in collaboration with the UV absorber, the detrimental effects of ultraviolet radiation on the polyurethane material are mitigated, thereby augmenting its resistance to yellowing. This study scrutinizes the positioning of UV absorber addition, the quantity of UV absorber, and the molar ratio of 1,4-butanediol to 2-hydroxyethyl disulfide, characterizing the functional groups of polyurethane through infrared and Raman spectroscopy. It is observed that the successful preparation of yellowing-resistant polyurethane is achieved, and evaluations on the modified polyurethane through color difference, tensile, and centrifugal tests reveal that the optimal yellowing resistance is attained by adding a UV absorber at a mass fraction of 1% to 3% prior to chain extension, resulting in a color change grade of 2, denoting slight discoloration. Simultaneously, the other properties of polyurethane exhibit relative stability. Notably, when the molar ratio of 1,4-butanediol to 2-hydroxyethyl disulfide is 3:2, the overall performance of the polyurethane remains stable, with exceptional yellowing resistance capabilities attaining a color change grade of 2.

## 1. Introduction

Polyurethanes are synthesized via polyaddition of diisocyanates or polyisocyanates with diols or polyols, consisting of soft and hard chain segments [1,2,3,4]. They exhibit excellent mechanical properties, thermal stability, and weatherability. Compared to metallic materials, polyurethanes possess advantages of light weight, good processability, low density, and high production capacity, and thus have been extensively utilized in application areas including leather coating, paints, and adhesives. However, waterborne polyurethanes are susceptible to yellowing under environmental exposure during production and service. Yellowing impairs product aesthetics, evidenced by the discoloration of ubiquitous transparent cell phone cases after some duration of use [5,6,7,8,9,10,11]. In daily life scenarios, white polyurethane shoe materials and some rubber polymer coatings also inevitably undergo yellowing under sunlight irradiation, severely hindering the manufacturing and use of waterborne polyurethanes in hot and tropical climates. Therefore, enhancing the yellowing resistance of waterborne polyurethanes is of great necessity [10,12,13,14,15].

Current approaches for improving polyurethane resistance to yellowing primarily focus on adjusting the hard/soft segment ratio and altering the synthetic building blocks. Strategies based on adding organic additives and nanomaterial modification have also been explored [15,16,17,18,19].

Aromatic isocyanates are routinely employed for conventional polyurethane preparation. Common aromatic diisocyanates include TDI and MDI, which tend to yield quinoid structures upon UV exposure and thus cause yellowing [19,20,21,22]. To mitigate this propensity, aliphatic diisocyanates such as HDI without aromatic moieties can be used as replacements [23,24,25,26]. Another indispensable component for polyurethane synthesis is polyol, commonly polyether and polyester types. Due to abundant unsaturated bonds, polyether, polyester, and polycarbonate polyols are vulnerable to oxidation into carboxylic acids and aldehydes/ketones when exposed to environmental elements. Subsequent catalyzed oxidation leads to chromophoric groups that accelerate polyurethane yellowing. Among different polyols, polyester exhibits better resistance to thermal-oxidative degradation relative to polyether counterparts, owing to the higher thermal stability of carbonyl groups compared to prone-to-oxidation ether linkages on the α-carbon [27]. The yellowing of polyurethanes arises from the combined effects of multiple factors, with light exposure being the most prominent contributor. Adding UV stabilizers during polyurethane preparation helps to curb yellowing, including light stabilizers and UV screening agents. UV absorbers constitute a major category of useful light stabilizers (Figure 1).

UV absorbers themselves can absorb incident UV radiation and convert it into benign thermal energy with no environmental impairment. Their working mechanism entails excited-state intramolecular proton transfer facilitated by UV irradiation, which cleaves the closed-shell intramolecular hydrogen bonds and generates unstable ketonic forms that relax to the ground state with photoenergy dissipated as heat, thereby alleviating UV-induced effects on the material system and decelerating yellowing [28,29,30]. Currently, common UV absorbers are added to polyurethane emulsions through physical blending, a method that not only requires a high concentration of UV absorbers, reaching 1 g/100 mL, but also has a significant impact on the stability of the emulsion. In this study, we selected UV–320, which possesses strong UV absorption capability, low toxicity, high stability, low volatility, and hydroxyl groups, as the UV absorber. We chemically grafted UV–320 onto the polyurethane chain segment. This approach not only requires a smaller quantity, with a concentration of just 1 g/200 mL, but also minimally affects the emulsion properties of the polyurethane. Furthermore, due to the chemical grafting of the UV absorber onto the polyurethane, and its good compatibility with polyurethane, the location of grafting the UV absorber onto the polyurethane chain segment has a minimal effect on the emulsion properties. Current approaches to curb yellowing also encompass adding relevant agents as stabilizers. For instance, He et al. [31] physically blended SiO_2_ nanoparticles with the aromatic diisocyanate MDI. Results showed that incorporation of 0.23 wt% SiO_2_ yielded a polyurethane emulsion with zero UV transmittance. This arises from the potent UV absorption capacity and infrared reflection properties of SiO_2_ nanoparticles. Moreover, insufficient surface coordination endows SiO_2_ nanoparticles with enhanced reactivity to adsorb certain chromophoric species, thus reducing photo-bleaching under UV irradiation. During film formation, a network structure can be established with SiO_2_ nanoparticles to markedly enhance coating resistance to degradation and corrosion [32]. Drawbacks of this method include poor compatibility between the inorganic nanomaterial and the organic polyurethane matrix, leading to emulsion instability and deterioration of coating mechanical performance.

A characteristic of disulfide bonds lies in their powerful dynamic reversibility, which results in an extremely low activation energy and the ability to break and reform multiple times, thus achieving more efficient structural adjustments. Due to the dynamic reversibility of disulfide bonds, under ultraviolet irradiation, these bonds continuously break and recombine, transforming the energy of ultraviolet light into thermal energy release. As the cause of yellowing is primarily the ability of ultraviolet radiation to excite chemical bonds within polyurethane materials, leading to bond cleavage and recombination reactions, resulting in structural changes and yellowing of the polyurethane, this study employs a synergistic approach involving the utilization of ultraviolet-absorbing polyurethane and disulfide compounds to convert absorbed ultraviolet light into thermal energy release, thereby mitigating the impact of ultraviolet radiation on polyurethane. By introducing the dynamic reversible substance 2-hydroxyethyldisulfide into the polyurethane synthesis process, which contains a disulfide compound with hydroxyl groups that readily react with isocyanate groups, it is utilized in the chain extension stage of polyurethane synthesis, introducing it into the polyurethane chain segment. Furthermore, due to its minimal impact on the color of polyurethane films, 2-hydroxyethyldisulfide is selected as the site for introducing disulfide bonds. UV–320, which contains hydroxyl groups, is also introduced into the polyurethane chain segment due to its property of readily reacting with isocyanate groups. However, the different order of addition will impact the molecular weight and structure of the polyurethane. Therefore, this study investigates the effects of varying the dosage of the UV absorber, the molar ratio of 2-hydroxyethyldisulfide to 1,4-butanediol on the yellowing resistance of polyurethane, and characterizes and tests the modified polyurethane through infrared spectroscopy, color difference, and tensile performance.

## 2. Results and Discussion

### 2.1. Fourier Transform Infrared Spectroscopy and Raman Spectroscopic Analysis

From Figure 2, it is discernible that the spectral domain ranging from 2260 to 2280 cm^−1^ is attributed to the asymmetric stretching vibrations of the –NCO group, which serves as the most efficacious characteristic peak for the identification of isocyanate moieties. The absence of such peaks within this interval in the graph permits the inference that the isocyanate groups in the raw materials have been entirely consumed in the reaction. Proximate to 1250 cm^−1^, a distinctive peak corresponding to the C–O–C bond is observed, while the presence of the carbonyl group, C=O, is denoted by a characteristic peak at approximately 1650 cm^−1^. Absorption peaks associated with phenyl rings and their derivatives typically manifest near 1570 cm^−1^ and 860 cm^−1^; within the graph, phenyl ring characteristic peaks are discernible in the vicinity of 1506 cm^−1^ and 900 cm^−1^. In the precursors used for synthesizing pure polyurethane, phenyl rings are conspicuously absent; however, the phenyl rings incorporated within the ultraviolet light absorber UV–320, which is added to the modified polyurethane, are persistent throughout the reaction, indicating that the ultraviolet light absorber is implicated in the reaction.

As discerned from Figure 3, it is evident that in the vicinity of 500 cm^−1^, the control group exhibits no significant peak values, whereas a marked peak prominence is observable in the 1% sample in comparison to the control, leading to the conclusion that disulfide linkages have been introduced into the prepolymer. Around 3000 cm^−1^, a pronounced characteristic peak is discernible, suggestive of the presence of molecules bearing aromatic constituents.

### 2.2. Emulsion Viscosity

Figure 4 depicts the relationship between varying UV–320 content and viscosity. Compared to the blank group, the modified emulsions exhibit increased viscosity across all gradients, save for the 1% sample, which shows little deviation. As ultraviolet absorbent content escalates, the resultant decline in molecular chain length lowers molecular weight, thereby diminishing molecular dimensions and augmenting viscosity [33]. Prior to the first chain extension, escalating ultraviolet absorbent dosage gradually elevates emulsion viscosity. A likely explanation is that over the reaction process, the escalation in ultraviolet absorbent content likewise increases dispersed ultraviolet absorbent in the polyurethane emulsion. Owing to the relatively high molecular weight of the ultraviolet absorbent, viscosity correspondingly increases, beyond other factors. Barring the 7% sample, the viscosity of the other mass fraction emulsions exceeds that prior to the first chain extension, potentially because the delayed addition time leads to incomplete reaction and residual UV–320, heightening emulsion viscosity. The viscosity of the 7% sample is lower than prior to the first chain extension, possibly due to the greater acetone volume added over the reaction process relative to the other gradients, reducing viscosity.

### 2.3. Emulsion Acid-Alkali Resistance

Upon perusal of Table 1, Table 2 and Table 3, one observes that the pH values of the control group, the emulsions with varying concentrations of ultraviolet absorbers, and those with different molar ratios of polyurethane consistently approximate a neutral pH of around 7. In the execution of acid and alkali resistance experiments, the introduction of various concentrations of hydrochloric acid and sodium hydroxide into the emulsions failed to precipitate any phenomena indicative of emulsion breakdown. When contrasted with the control group, the modified emulsions exhibited an acid-alkali resistance essentially commensurate with that of the unmodified group, suggesting that the incorporation of the two chemical agents did not compromise the emulsion’s resistance to acid and alkali. Divergent from anionic and cationic aqueous polyurethanes, nonionic aqueous polyurethanes, devoid of hydrophilic anionic and cationic groups, consequently demonstrate superior acid and alkali resistance capabilities relative to their anionic and cationic aqueous polyurethane counterparts [34,35,36,37].

### 2.4. Emulsion Stability

As evidenced in Figure 5 and Table 4, at a concentration range of 5% to 9%, the substantial steric hindrance presented by the ultraviolet absorbers impedes their reaction, which, as the concentration of the ultraviolet absorber increases, results in larger particles of the absorber within the dispersed polyurethane emulsion and a consequent distribution that is not uniform, thereby diminishing the stability. When the concentration of the ultraviolet absorber is between 1% and 3%, the emulsion is relatively stable with negligible precipitation. Comparing the stability of the emulsion prior to the first chain extension with the stability after the addition of water, the former exhibits marginally superior stability. This is due to a higher content of isocyanate groups before the first chain extension, which allows for a more complete reaction with the ultraviolet absorber, thus enhancing stability. Controlling the concentration of the ultraviolet absorber at 2% and manipulating the molar ratio of 2-hydroxyethyl disulfide to 1,4-butanediol, it was observed that as the proportion of 2-hydroxyethyl disulfide increased to 60%, the emulsion tended towards instability. This is attributed to the relatively weak intermolecular forces of disulfide bonds, which are prone to cleavage, leading to increased instability of the emulsion with higher quantities of 2-hydroxyethyl disulfide.

### 2.5. Emulsion Particle Size

The particulate dimensions of the polyurethane emulsions were ascertained utilizing a BT-9300H laser particle size analyzer. Concurrently with the acquisition of the polyurethane emulsion globule diameters, the specific surface area of the polyurethane emulsions was obtained. As depicted by Figure 6 and Figure 7, all gradient-specific surface areas except the 1% sample exceed the blank, potentially attributable to diminished emulsion particle size from ultraviolet absorbent addition, while the negligible deviation of the 1% sample arises due to the lower UV–320 volume. As the UV–320 mass fraction increases, the emulsion-specific surface area displays an overall growth trend. Specific surface area relates to particle size, with finer particles begetting a larger area. Within a defined concentration range, smaller average particle size concentrates the emulsion while larger dimensions dilute it. The test emulsions evidenced declining particle size and a rising concentration from 1–9%, with particle dimensions closely tied to emulsion state. With water as the carrier, a minimal specific surface area renders the emulsion transparent and faintly blue, transitioning to translucent white, then opaque white with a climbing area.

### 2.6. UV-Visible Absorption Spectroscopy

In this experiment, the Cary 300 UV-visible spectrophotometer was used to conduct the sample testing. Three milliliters of the polyurethane emulsion was taken in a cuvette, with a scanning interval of 5 nm and a wavelength range of 800~200 nm. As depicted in Figure 8, the blank sample solely displays an absorption peak at 230 nm. Comparatively, the addition of a disulfide introduces a 400 nm peak, while the independent and combined incorporation of an ultraviolet absorbent and disulfide manifest high wavenumber absorption peaks absent in the blank sample. By integrating the analysis of color difference results, it can be observed that the addition of disulfides enhances the yellowing resistance capability of polyurethane, attributed to the color differences that manifest within the ultraviolet-visible light spectrum. Aside from the 230 nm wavelength that damages polyurethane, other wavelengths severely degrade the material. The introduction of a disulfide compound enables reversible disulfide bond alterations to absorb higher wavelength visible light at 400 nm as well, improving yellowing resistance.

### 2.7. Chromatic Aberration

From Figure 9, it is evident that upon contrasting the pre-crosslinking and pre-water addition samples with the blank group, it is observed that under 300 W ultraviolet lamp exposure for 3 h and 6 h, the total color difference (ΔE) of the blank group is significantly higher than the other five gradients. According to the color change rating range established by GB/T 1766-2008 [38], the blank group undergoes distinct color variation. In the scenario where the UV–320 ultraviolet absorber is introduced before the initial crosslinking, the 1% mass fraction demonstrates the most favorable outcome, with ΔE = 3.2, signaling a color change rating of 2, with only a slight chromatic shift. The total color differences from 1% to 9% exhibit an increasing trend but all remain below the blank group. In the scheme where the UV–320 ultraviolet absorber is added before the water inclusion, the sample with a 7% mass fraction displays minimal total color difference, ΔE = 2.6, and a color change rating of 1, evidencing only very slight color variation. Compared to the blank group, the color difference within the 1% to 9% range shows minimal variation, indicating a relatively robust resistance to yellowing.

Figure 10 illustrates an overall decreasing then increasing total color difference trend when fixing ultraviolet absorbent quantity and altering the 1,4-butanediol to 2-hydroxyethyl disulfide molar ratio. The 3/5 disulfide proportion fares slightly better than other samples. Barring this, the residual four sample color change magnitudes are smaller than the blank, with only slight alteration following 9-h irradiation. Liu et al. [39] conducted research on the impact of ultraviolet absorbers on the UV resistance of PUR. The study revealed that the system with added ultraviolet absorbers was more stable compared to the system without such additives, as evidenced by the increased retention of tensile strength and elongation at break. Under conditions of xenon lamp irradiation for 6 h, the total color difference ΔE reached 8 with the addition of 2-(2H-benzotriazol-2-yl)-4,6-ditertpentylphenol to tolylene diisocyanate-based polyurethane, 7 with the addition of bis(1,2,2,6,6-pentamethyl-4-piperidinyl) sebacate, and 6 with the addition of 3-(3,5-di-tert-butyl-4-hydroxyphenyl) propionic acid octyl ester. The total color difference ΔE in this study was only 3, corresponding to a color change level of 2, indicating only slight yellowing.

### 2.8. Thermal Gravimetric Analysis

From the analysis of Figure 11, it is ascertained that, subsequent to modification, the thermogravimetric (TG) curve of the polyurethane film and the derivative thermogravimetry (DTG) curve exhibit no marked divergence in trend when compared to the control group, essentially superimposing upon one another. This observation leads to the deduction that the introduction of disulfide bonds and an ultraviolet absorber does not materially alter the thermal stability of the polyurethane.

### 2.9. Tensile Properties

As discerned from Figure 12, the polyurethane films prepared under both schemes achieve their maximal elongation at break and tensile strength at a 5% concentration, indicating that at this mass fraction, the samples exhibit the greatest tensile strength and elongation at break, indicating superior flexibility and elasticity, as well as relatively enhanced toughness compared to other gradations. In comparison with the control group, both schemes demonstrate an improved tensile strength. This enhancement is likely attributable to the presence of phenyl rings within the ultraviolet absorber, which, due to their rigidity, foster a trend of increasing tensile strength with the increase of the UV absorber concentration. Within both schemes, the 1% and 3% concentrations appear comparatively stable.

Figure 13 reveals that with the incremental addition of BDO (1,4-butanediol), the tensile strength of the samples rises, reaching a zenith, beyond which it begins to wane. This may be due to the relatively low molecular weight and short chain segments of BDO, which can effectively enhance intermolecular interactions, thereby augmenting the hardness, fracture strength, and modulus of the polyurethane upon increasing its concentration. When juxtaposed with the control group, both tensile performance and elongation at break demonstrate varying degrees of improvement, signifying that the mechanical properties of the modified polyurethane are fortified.

### 2.10. Dynamic Mechanical Analysis

Inspection of Figure 14 discloses that the glass transition temperature (Tg) of the control group is proximate to −48 °C, while the modified emulsions exhibit a Tg not significantly divergent from that of the control. When the polyurethane is above its glass transition temperature, it assumes an elastomeric state. The lower the glass transition temperature, the greater the flexibility of the polymer chains. The Tg of the modified polyurethane emulsions is marginally higher than that of the control group, a consequence of the added modifiers contributing to an increase in the molecular weight of the polymer, thereby elevating the glass transition temperature.

### 2.11. Mechanism of Yellowing Resistance

Luminescent solar radiation impinges upon the polyurethane surface, where a portion of the ultraviolet spectrum is reflected back into the atmosphere by the polyurethane, whilst the remainder is refracted through the interface and penetrates the interior of the polyurethane material. A fraction of the ultraviolet rays induces the cleavage of disulfide bonds, generating unstable sulfur radicals. The ruptured sulfur radicals subsequently recombine to form novel disulfide linkages. Throughout this ongoing cycle of disintegration and reconstitution, the energy of the ultraviolet rays is transmuted into thermal energy and dissipated, thereby mitigating the deleterious effects of ultraviolet radiation on the polyurethane substrate. The residual ultraviolet rays, refracted into the interior of the polyurethane, provoke the intramolecular hydrogen bonds of the UV–320 molecules, which are in a closed state, to reach an excited state. Through proton transfer, the intramolecular hydrogen bonds are cleaved, resulting in an unstable keto configuration that reverts to the ground state, concurrently converting photonic energy into thermal energy that is also dissipated. The synergistic interaction between disulfide bonds and UV–320 facilitates the conversion of ultraviolet energy across varying wavelengths into thermal energy, thus minimizing the degradation of the polyurethane material by ultraviolet light and achieving the effect of resistance to yellowing (Figure 15).

## 3. Materials and Methods

### 3.1. Reagents and Instrumentation

2-Hydroxyethyl disulfide, chemically pure, Shanghai Macklin Biochemical Co., Ltd. (Shanghai, China); UV absorber UV–320, chemically pure, Shanghai, China, Shanghai Macklin Biochemical Co., Ltd.; dibutyltin dilaurate, chemically pure, Angene International Limited (Beijing, China); polytetramethylene ether glycol (PTMEG), MW 2000, Shanghai, China, Shanghai Macklin Biochemical Co., Ltd.; polytetramethylene ether glycol (PTMEG), MW 1000, Shanghai, China, Shanghai Macklin Biochemical Co., Ltd.; polypropylene glycol (PPG), MW 1000, Shanghai, China, Shanghai Macklin Biochemical Co., Ltd.; isophorone diisocyanate (IPDI), chemically pure, Shanghai, China, Shanghai Macklin Biochemical Co., Ltd.; acetone, analytically pure, Xiyu Chemical Technology Co., Ltd. (Shandong, China); 1,4-Butanediol (BDO), chemically pure, Tianjin Bodie Chemical Co., Ltd. (Tianjin, China); polyethylene glycol (PEG), chemically pure, Shanghai, China, Shanghai Macklin Biochemical Co., Ltd.; polyethylene glycol monomethyl ether (MPEG), chemically pure, Shanghai, China, Shanghai Macklin Biochemical Co., Ltd.; polyethylene glycol (PEG), chemically pure, Shanghai, China, Shanghai Macklin Biochemical Co., Ltd.; hydrogenated diphenylmethane diisocyanate (HMDI), chemically pure, Shanghai, China, Shanghai Macklin Biochemical Co., Ltd.

Oil bath with digital display for temperature control, single hole, Longkou Xianke Instrument Co., Ltd. (Shandong, China); precision-enhanced stirrer, JJ-1, Guohua Electric Appliance Co., Ltd. (Henan, China); constant temperature and humidity incubator, JWS-150, Shanghai Shanlian Equipment Co., Ltd. (Shanghai, China); electronic balance, AR3130, Aohaus Corporation (Shanghai, China); muffle furnace, SX2-4-10, Longkou Xianke Instrument Co., Ltd. (Shandong, China) of Shandong Province; electronic balance, AR3130 (Shanghai, China), Aohaus Corporation; universal color difference meter, JZ-300, Shenzhen Jinzhun Instrument Equipment Co., Ltd. (Shenzhen, China); yellowing resistance tester (light bulb type), HZ-3017, Dongguan Dexiang Instrument Co., Ltd. (Guangdong, China); digital display viscometer with microcomputer control, NDJ-5S, Shanghai Yutong Instrument and Meter Factory (Shanghai, China); blast drying oven, DHG-9080, Changzhou Jintan Precision Instrument Manufacturing Co., Ltd. (Jiangsu, China); multi-purpose vacuum pump with circulating water, SHZ-D(III), Gongyi Honghua Instrument Equipment Industry and Trade Co., Ltd. (Henan, China); vacuum drying oven, DZF-6020, Shanghai Xinmiao Medical Device Manufacturing Co., Ltd. (Shanghai, China); computer-controlled electronic universal testing machine, UTM2502HB, Shenzhen Sansi Horizontal and Vertical Technology Co., Ltd. (Shenzhen, China); cycle dispersing device, BT-600, Dandong Baite Instrument Co., Ltd. (Liaoning, China).

### 3.2. Experimental Method

#### 3.2.1. Synthesis of Yellowing-Resistant Polyurethane

A quantity of 8 g of PTMEG (MW = 1000), 8 g of PTMEG (MW = 2000), 4 g of IPDI, 8 g of HMDI, and 0.1 g of dibutyltin dilaurate were added to a three-necked flask. A stirrer was inserted into the flask, which was then fixed to an iron stand with the stirrer immobilized. The stirring rate was adjusted to 250 rpm and stirred at 75 °C for 1 h. The oil bath temperature was increased to 80 °C before incorporating 6.4 g of polyethylene glycol, allowing reaction for 1 h. Then, 0.22 g of BDO and 4 g of MPEG were introduced and reacted for another 1 h, closely monitoring the viscosity throughout. Acetone was added in small, multiple timely batches to reduce viscosity when substantial increases were observed. After lowering the oil bath to room temperature, water was added while maintaining 2000 rpm for 30 min reaction. The speed was reduced back down to 250 rpm as 30% ethylenediamine (EDA) aqueous solution was poured in for the second chain extension, proceeding for approximately 60–90 min until most of the foam subsided to finish the reaction.

#### 3.2.2. Modification for Yellowing Resistance

In the primary chain extension process, 2-hydroxyethyl disulfide and BDO were employed, while the ultraviolet absorber was applied at different stages, namely prior to the pre-polymerization secondary chain extension and before the addition of water, in order to identify the optimal position for the utilization of the ultraviolet absorber. Subsequently, the ultraviolet absorber was implemented at the identified optimal position (Figure 16).
(1)Investigating the effect of changing the mass of the UV absorber under a constant molar ratio of 2-hydroxyethyldisulfide to 1,4-butanediol to determine the appropriate mass: The UV absorber is divided into five gradients of 1%, 3%, 5%, 7%, and 9% by mass, and added either before the first chain extension or before emulsification with water. The different positions and concentrations of the UV absorber will influence the length of the polyurethane chain segment, as well as the molecular weight and molecular structure arrangement, thereby affecting the stability of the polyurethane emulsion and the mechanical and thermal properties of the film. This experiment aims to determine the optimal position and concentration for the addition of the UV absorber (Table 5).(2)Investigating the effect of adding the same UV absorber under different molar ratios of 1,4-butanediol to 2-hydroxyethyldisulfide, with ratios of 0:5, 1:4, 2:3, 3:2, 4:1, and 5:0, in six parallel experiments: by using 2-hydroxyethyldisulfide to partially replace BDO in the chain extension, this experiment aims to determine the optimal ratio of 2-hydroxyethyldisulfide to BDO (Table 6).

Initially, the polyurethane prepolymer is formed through the reaction between polyols and isocyanate groups. As non-ionic waterborne polyurethane does not generate ions during the synthesis process and exhibits poor hydrophilicity, the introduction of PEG is employed to incorporate hydrophilic substances and enhance its hydrophilicity, thereby preparing the polyurethane for emulsification with water. Subsequently, UV–320, a UV absorber with inherent hydroxyl groups, is added. This UV absorber possesses strong UV absorption capacity, low toxicity, high stability, and low volatility, effectively shielding the polyurethane from UV exposure and reducing UV-induced damage to the polyurethane matrix. By leveraging its inherent hydroxyl groups, UV–320 undergoes a reaction with the remaining isocyanate groups to introduce it into the polyurethane chain segment. Furthermore, 2-hydroxyethyl disulfide, possessing two hydroxyl groups, meets the requirements for a chain extender. By reacting the dual hydroxyl groups of 2-hydroxyethyl disulfide with the isocyanate groups at both ends of the chain segment using BDO, disulfide bonds are incorporated into the polyurethane chain segment, further elongating the polyurethane chain segment. Subsequent addition of modified PEG introduces additional hydrophilic substances, followed by the introduction of water for emulsification under mechanical shearing. Finally, ethylenediamine is added for secondary chain extension, resulting in the formation of the non-ionic waterborne polyurethane emulsion.

### 3.3. Characterization and Performance Evaluation

#### 3.3.1. Fourier Transform Infrared (FT-IR) and Raman Spectroscopic Characterization

The 1 mm thick polyurethane film was sliced and fixed on the sample holder. The thermosetting polyurethane film underwent infrared spectroscopy scanning in the wavelength range of 400~4000 cm^−1^ using the ATR mode on the Thermo Scientific Nicolet iS20 spectrometer (Waltham, MA, USA). The thermosetting polyurethane film underwent Raman spectroscopy analysis using the Senterra confocal Raman microscope (BRUKER, Bremen, Germany), with a laser light source of 532 nm and a spectral range of 400~4000 cm^−1^, to verify the successful conjugation of ultraviolet stabilizers and disulfide bonds.

#### 3.3.2. Emulsion Viscosity

The viscosity of waterborne polyurethane emulsion is one of the performance indicators for waterborne polyurethane products, and viscosity is positively correlated with solid content. Different usage scenarios impose varying requirements on the viscosity of the polyurethane emulsion. In this experiment, the NDJ-5S viscometer was utilized to measure the emulsion viscosity. A volume of 25 mL of the solution to be tested was placed in a beaker. Prior to the measurement, the viscosity of the fluid to be measured was evaluated, and then an appropriate spindle and speed were selected from the range table. When an approximate viscosity value cannot be estimated, the maximum value is used, and the rotor is selected from small to large and from slow to high speed. The temperature of the emulsion to be tested was maintained at a consistent level. Subsequently, 20–25 mL of the liquid to be tested was added to the outer test cylinder. When the machine is in the standby interface, the measurement interface is accessed by pressing the OK button, and the corresponding rotor model and speed are selected to commence the measurement. To ensure accuracy, the percentage range reading during measurement should be controlled between 10% and 90%.

#### 3.3.3. pH Stability Characterization

The acid and alkali resistance of emulsions has a significant impact on the range and scenarios of polyurethane usage, with better acid and alkali resistance expanding the utility scope of polyurethane and increasing its research value. In this experiment, the pH testing method was employed: 1 mL of polyurethane emulsion was measured into a test tube using a pipette, with three identical sets. Pre-calibrated sodium hydroxide solution and hydrochloric acid solution were then added drop by drop using a dropper to the test tubes containing the polyurethane emulsion until the emulsion exhibited a demulsification phenomenon and produced flocculent precipitate, at which point the addition was ceased. The contents of the two test tubes were then thoroughly mixed by vibration, and clean glass rods were used to transfer drops of the emulsion from each test tube onto respective pH test papers for comparison to determine the emulsion’s resistance to acid and alkali. Clean test tubes were used to transfer blank samples for comparison with pH test papers to determine the original pH value of the emulsion.

#### 3.3.4. Emulsion Stability Evaluation

The stability of emulsions is one of the crucial performance indicators for waterborne polyurethane products, with better emulsion stability correlating to longer storage time. In this experiment, approximately 4 mL of uniform polyurethane emulsion was taken in a centrifuge tube and subjected to centrifugation at a speed of 5000 r/min for 8 min at room temperature. The centrifuged sample was then observed for any signs of layering or precipitation at the bottom.

#### 3.3.5. Particle Size Measurement

Particle size distribution is one of the key factors affecting emulsion stability. When the particle size in the emulsion is large or has a broad distribution range, the protection of the dispersed structure is compromised. Therefore, it is essential to employ appropriate emulsification methods and techniques to control the average particle size, size distribution range, and shape, ensuring that the emulsion particle size is suitable and uniform. In this experiment, a laser particle size analyzer was used to measure the particle size of the polyurethane emulsion. After shaking the sample, 10 mL of the sample was taken and added to a container containing about 300 mL of deionized water. Then, the sample to be tested was slowly dripped into the container until the measured concentration value on the instrument was in the range of 10~60 μg/mL. If the concentration data is less than 10 μg/mL, it indicates that the sample concentration in the container is too low and more sample should be added. Conversely, if the concentration data is greater than 60, more deionized water should be added. The test was completed about two minutes after sonication.

#### 3.3.6. UV-Visible Absorbance Spectroscopy

The ultraviolet-visible absorption spectrum test can provide absorbance values of different wavelengths for different polyurethane emulsions. In this experiment, the Cary 300 UV-visible spectrophotometer was used to conduct the sample testing. Three milliliters of the polyurethane emulsion was taken in a cuvette, with a scanning interval of 5 nm and a wavelength range of 800~200 nm.

#### 3.3.7. Color Difference Evaluation

Through color difference testing, the yellowing resistance performance of polyurethane films can be visually determined. In this experiment, the JZ-300 universal colorimeter was utilized. The film was cut into samples 50 mm × 50 mm and a thickness of approximately 1 mm. The modified samples and blank samples were placed together in a light bulb-type yellowing resistance test chamber. The samples were exposed to a 300 W ultraviolet lamp at a distance of 25 cm from the light source while maintaining a temperature of approximately 60 °C inside the chamber for 3 h and 6 h. Various parameters of the colorimeter were recorded and compared with the color change grades specified in GB/T 1766-2008 (Table 7) [38].

#### 3.3.8. Thermogravimetric Analysis

The thermal stability of the polyurethane samples was assessed using the TGA4000 thermogravimetric analyzer from PerkinElmer (Waltham, MA, USA). All samples were thoroughly dried for 24 h prior to testing. Each test involved weighing approximately 10 mg of polyurethane film samples, with an acceptable deviation of ±0.5 mg. The polyurethane film was then placed in a clean vessel and positioned within the instrument for testing. The testing environment parameters were adjusted, with the test gas atmosphere being nitrogen and the heating rate set at 10 K/min, ramping up from 300 °C to 700 °C.

#### 3.3.9. Tensile Properties

To evaluate mechanical strengths, tensile testing was conducted to obtain key quantifiable metrics (ultimate tensile strength and elongation at break). At 20 °C, dumbbell-shaped specimens with 0.5~1 mm thickness and 5 mm width were stretched to rupture at 50 mm/min test speed using a UTM2502HB computer-controlled electronic universal tester, while maximum strengths and elongations were documented.

#### 3.3.10. Dynamic Mechanical Analysis

Specimens were shaped into thin strips measuring approximately 3 mm in length, 5 mm in width, and 1 mm in thickness. Subsequently, the dynamic thermomechanical analysis (DMA) technique was employed on a TA Instruments Q800 (New Castle, DE, USA) with a tension membrane mode, utilizing a frequency of 1 Hz and a strain amplitude of 1%. The heating rate was set at 5 °C/min, spanning a temperature range from −100 °C to 150 °C.

## 4. Conclusions

In summation, we have integrated the ultraviolet absorbent UV–320 into the polyurethane prepolymerization reaction and introduced the dynamically reversible substance 2-hydroxyethyl disulfide as a chain extender in the synthetic process of polyurethane. The investigation encompassed the timing of addition for the ultraviolet absorbent, its proportional use, and the molar ratio between 1,4-butanediol and 2-hydroxyethyl disulfide. Characterization of the distinctive functional groups of polyurethane was conducted utilizing infrared spectroscopy and Raman spectroscopy. When the molar ratio of 1,4-butanediol to 2-hydroxyethyl disulfide stood at 3:2, the discoloration level was assessed at a grade of 2. The inclusion of 1% to 3% by mass fraction of the ultraviolet absorbent in the prepolymer, in conjunction with the addition of 2-hydroxyethyl disulfide during chain extension, resulted in only slight discoloration, amounting to a discoloration grade of 2, and exhibited substantial resistance to yellowing. Concurrently, the mechanical properties, thermal stability, and emulsion stability of the polyurethane remained relatively constant. This article introduces the ultraviolet absorbent into the main chain of polyurethane and leverages the potent dynamic reversibility of disulfide bonds, which possess extremely low activation energies and are capable of multiple cleavages and reconstructions. This synergistic action absorbs ultraviolet rays and enhances the resistance of polyurethane to yellowing. It paves the way for the broader application of polyurethane in fields such as construction, automotive, and leather industries, and offers novel insights for research on the lightfastness of polyurethane.

## Figures and Tables

**Figure 1 molecules-29-02099-f001:**
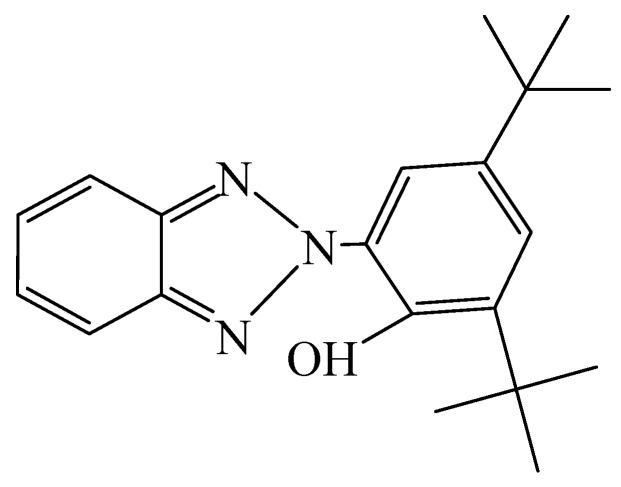
Chemical structure of UV absorber UV–320 with a benzotriazole motif.

**Figure 2 molecules-29-02099-f002:**
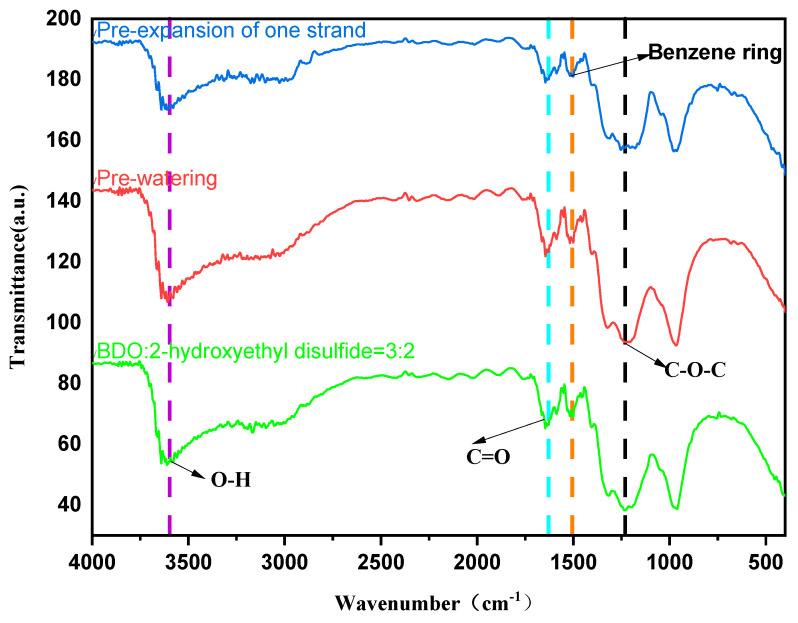
The infrared spectrum of UV–320 end-capping and the 2-hydroxyethyl disulfide chain extension (the position of UV–320 addition before primary chain extension and before water addition. The function of the various colored dashed lines is to facilitate the visual distinction of the presence of wave peaks at the current wave number).

**Figure 3 molecules-29-02099-f003:**
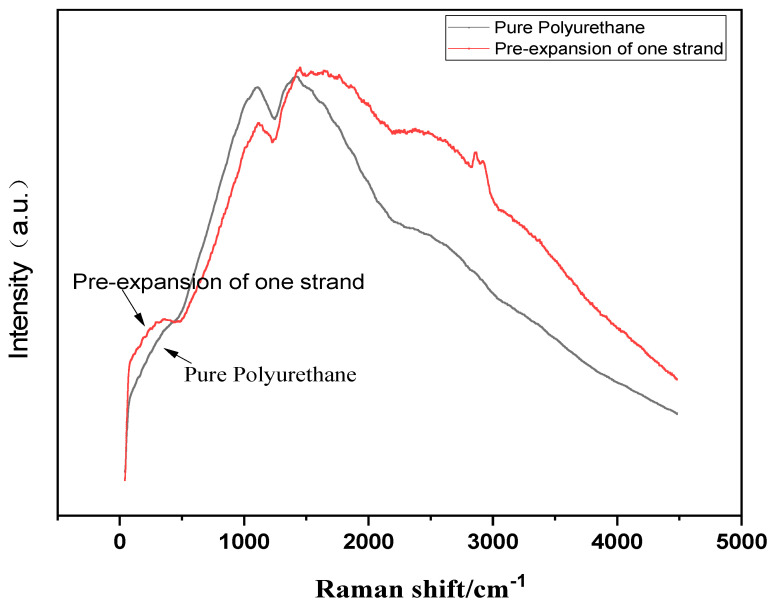
Raman spectrum of yellowing−resistant polyurethane (with a laser light source of 532 nm and a spectral range of 400~4000 cm^−1^).

**Figure 4 molecules-29-02099-f004:**
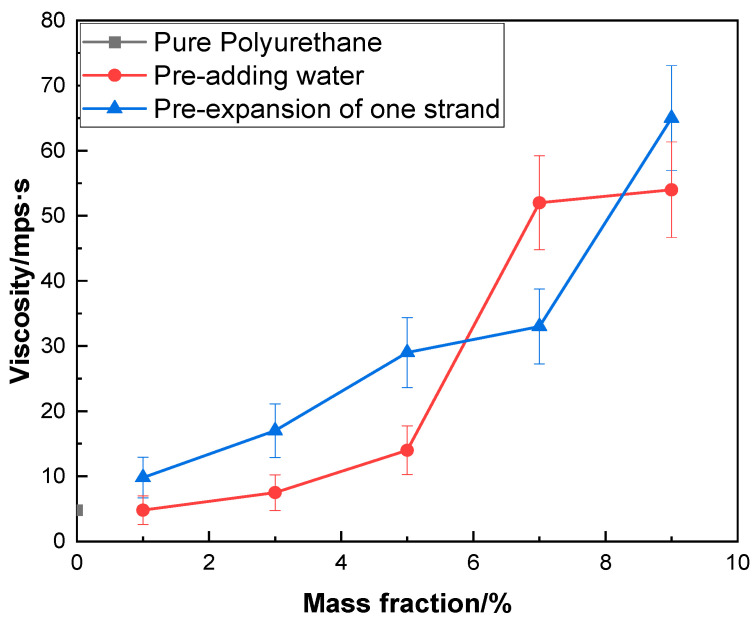
The correlation between different concentrations of UV–320 and viscosity.

**Figure 5 molecules-29-02099-f005:**
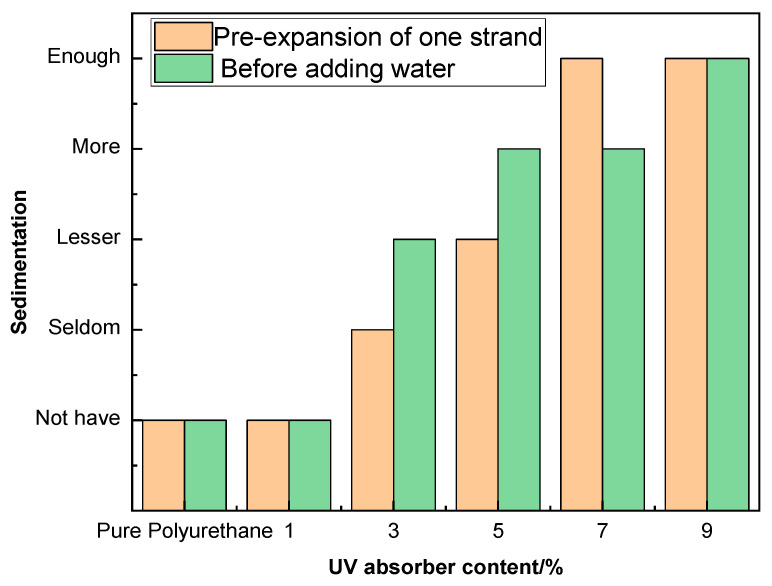
Precipitation of polyurethane emulsion with different UV–320 content. (Seldom: around 1/20 of a 5 mL centrifuge tube; Lesser: around 1/15 of a 5 mL centrifuge tube; More: around 1/10 of a 5 mL centrifuge tube; Enough: around 1/7 of a 5 mL centrifuge tube).

**Figure 6 molecules-29-02099-f006:**
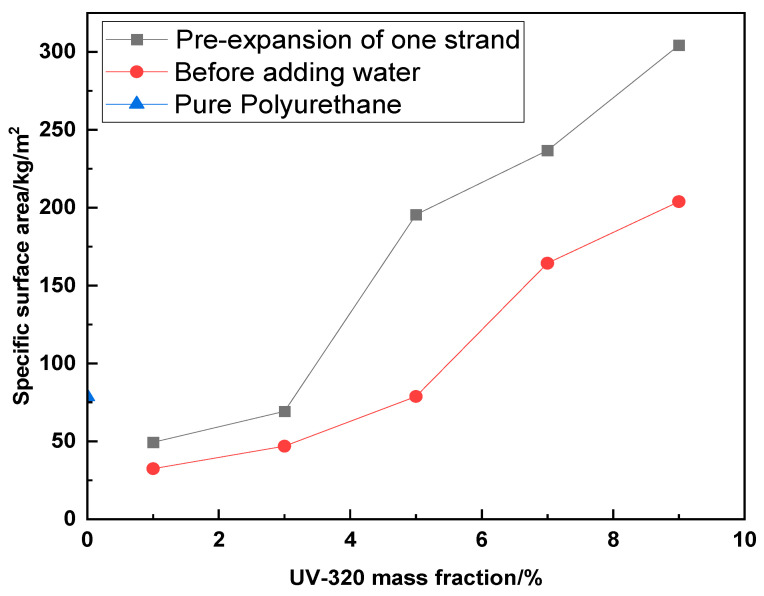
Relationship between different mass fractions and specific surface areas. (The data were obtained through measurements taken using the BT-9300H instrument in an environment set at 25 degrees Celsius and 60% humidity).

**Figure 7 molecules-29-02099-f007:**
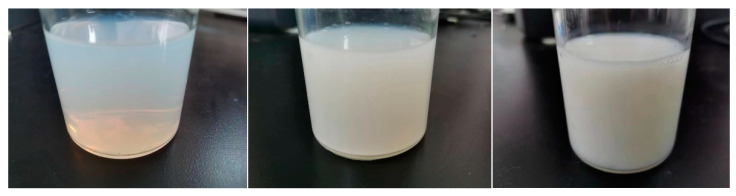
Polyurethane emulsion with UV absorber at 1%, 3%, and 7% (from left to right) before primary chain extension.

**Figure 8 molecules-29-02099-f008:**
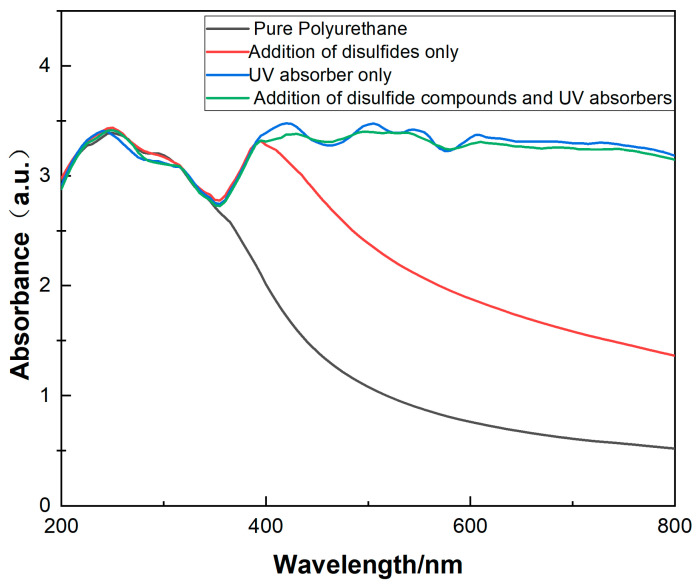
UV-Visible absorption spectrum of polyurethane with different materials. (Pure polyurethane: unmodified polyurethane; addition of disulfides only: polyurethane modified with a ratio of 3:2 of BDO to 2-hydroxyethyl disulfide; UV absorber only: polyurethane modified with a 3% content of UV absorber; addition of disulfide compounds and UV absorbers: polyurethane modified with a ratio of 3:2 of BDO to 2-hydroxyethyl disulfide along with a 3% content of UV absorber. For specific modification methods, please refer to Section 3.2.2).

**Figure 9 molecules-29-02099-f009:**
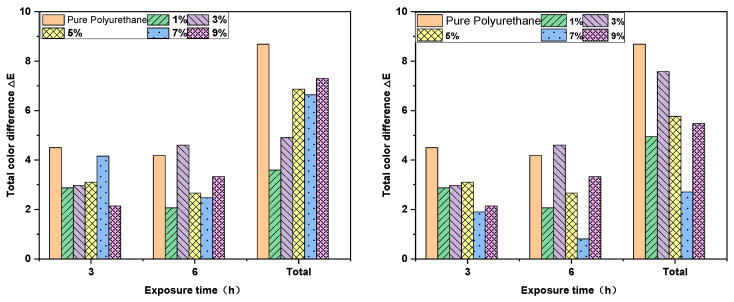
Relationship diagram between the irradiation time of polyurethane and the total color difference ΔE at different doses of UV–320 ((**left**): before primary chain extension, (**right**): before water addition).

**Figure 10 molecules-29-02099-f010:**
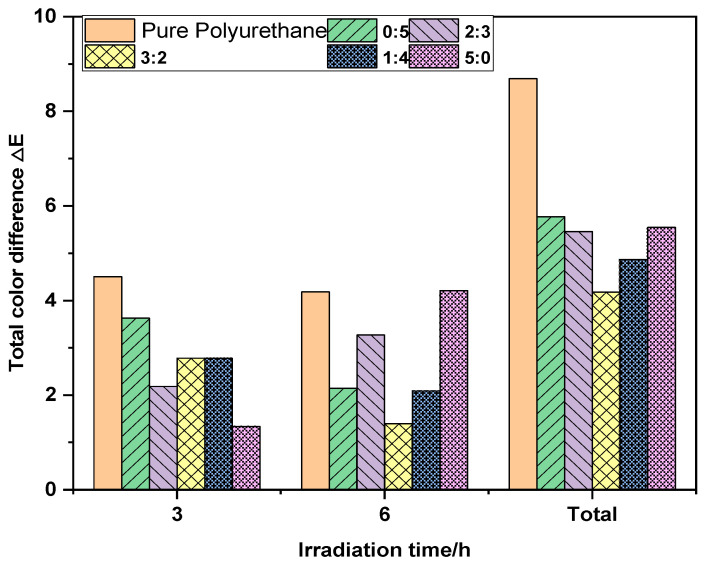
The correlation between the irradiation time and the total color difference ΔE beneath polyurethane films of varying molar ratios (1,4-butanediol: 2-hydroxyethyl disulfide).

**Figure 11 molecules-29-02099-f011:**
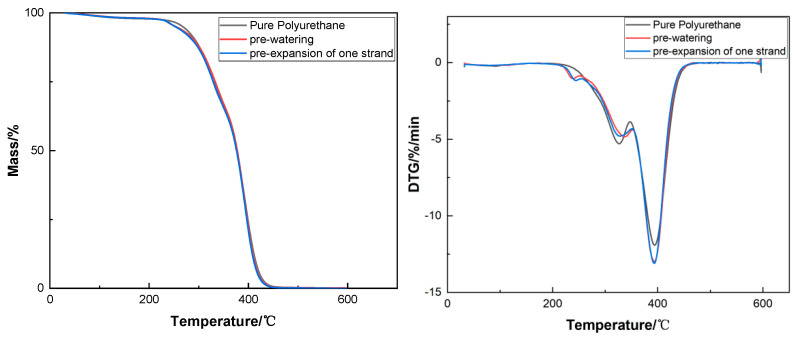
Thermogravimetric analysis (**left**) and derivative thermogravimetry (**right**) of polyurethane synergistically stabilized with UV–320 and 2-mercaptoethanol.

**Figure 12 molecules-29-02099-f012:**
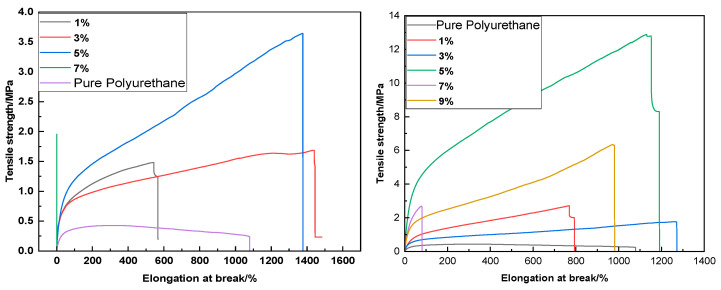
Influence of UV–320 on polyurethane film mechanical performance ((**left**): prior to first chain extension; (**right**): prior to water addition).

**Figure 13 molecules-29-02099-f013:**
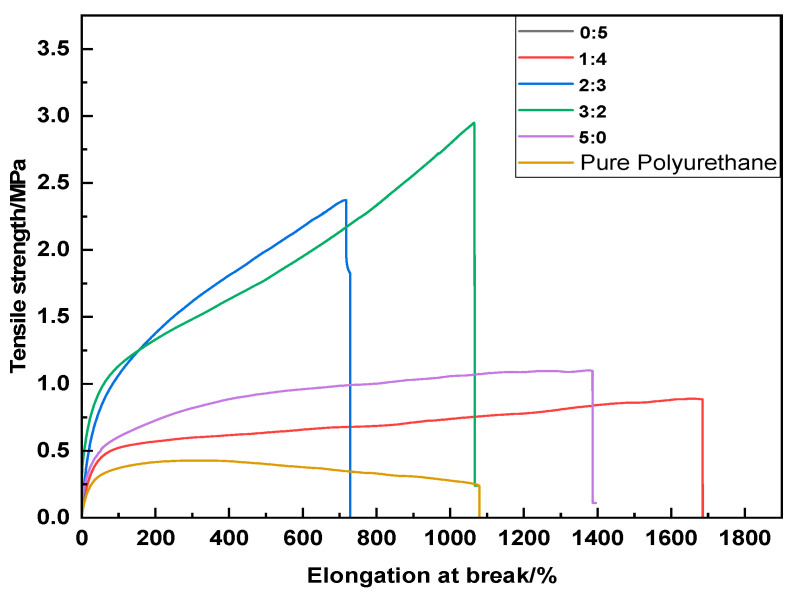
Influence of varying molar ratios of UV–320 and 2-mercaptoethanol on the mechanical properties of polyurethane films.

**Figure 14 molecules-29-02099-f014:**
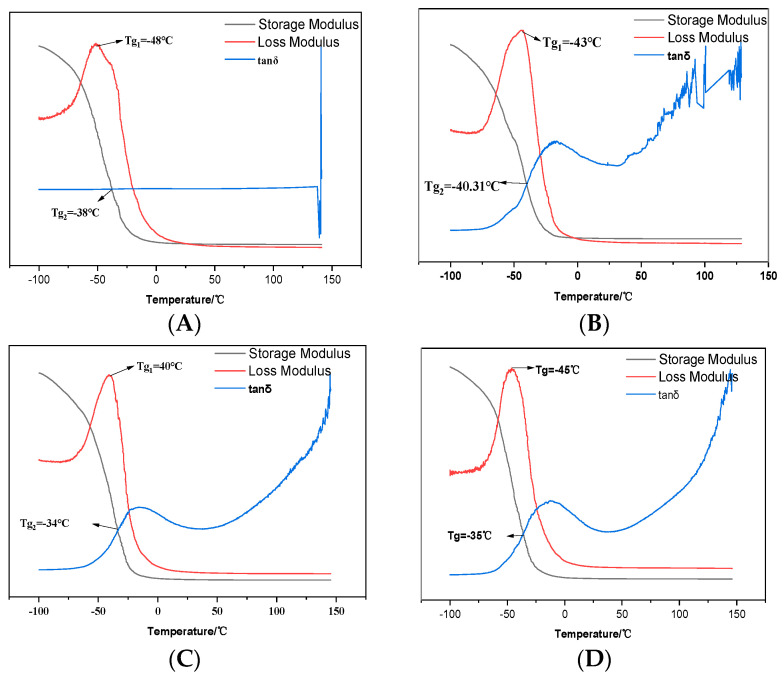
Dynamic thermomechanical analysis of polyurethane films derived from UV–320 and 2-mercaptoethanol ((**A**): blank; (**B**): before UV–320 and water added; (**C**): before first chain extension of UV–320; (**D**): molar ratio of 2-mercaptoethanol and BDO is 2:3).

**Figure 15 molecules-29-02099-f015:**
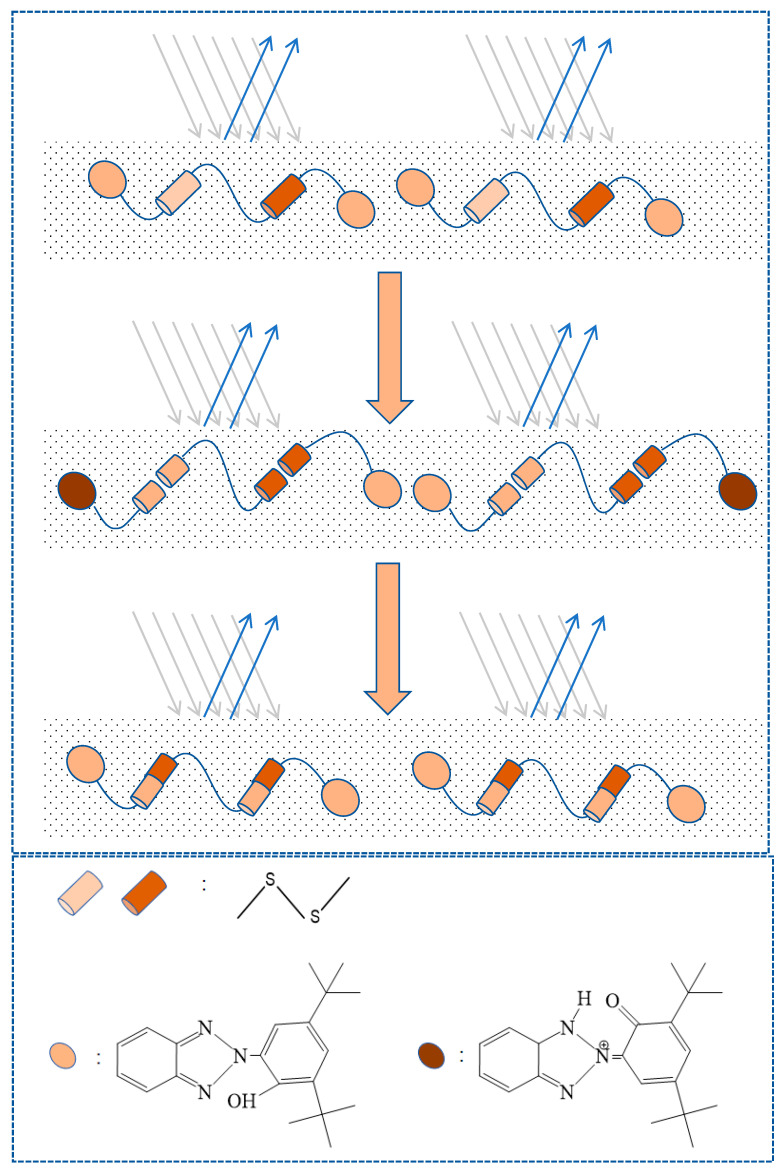
Mechanism of yellowing resistance. (The black line signifies the incidence of sunlight rays, while the blue line represents the light rays that undergo partial reflection).

**Figure 16 molecules-29-02099-f016:**
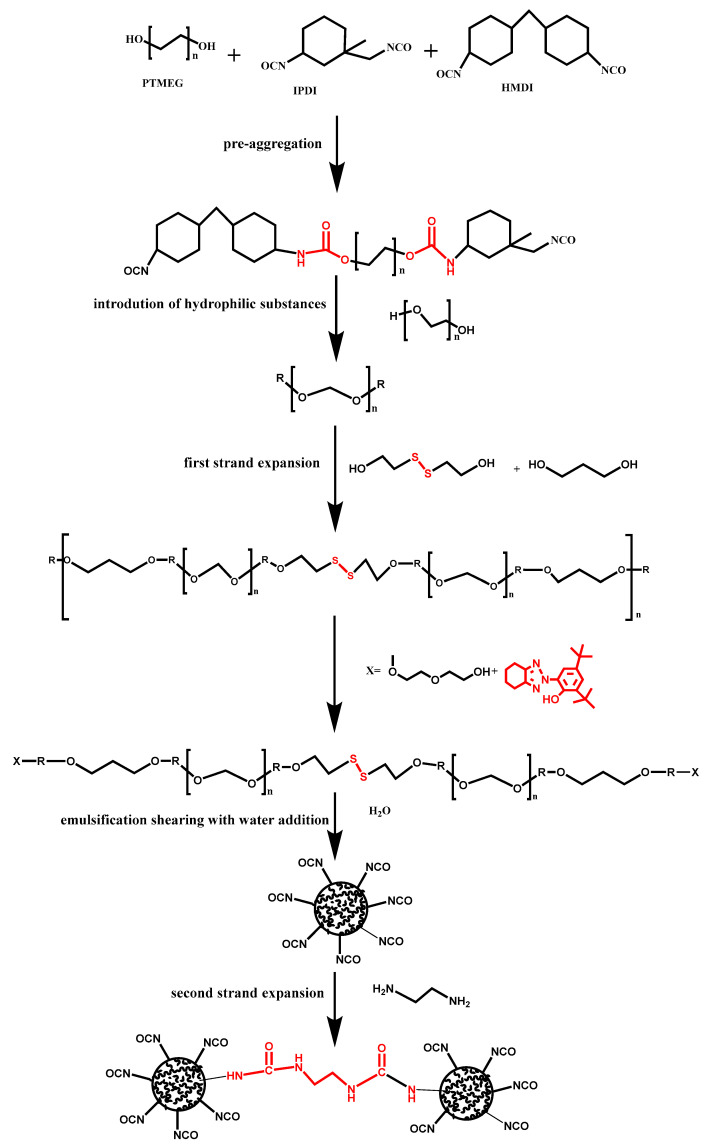
The reaction equation for the yellowing-resistant non-ionic waterborne polyurethane. The entire experimental procedure encompasses the following steps: pre-aggregation, introduction of hydrophilic substances, first strand expansion, emulsification shearing with water addition, and second strand expansion.

**Table 1 molecules-29-02099-t001:** The acid and alkali resistance of polyurethane emulsion at different molar ratios of BDO and 2-hydroxyethyl disulfide.

Mole Ratio	Acid pH Resistance	Neutral	Alkali pH Resistance
blank	1 ± 0.5 (stable)	7 ± 0.5	13 ± 0.5 (stable)
1:4	1 ± 0.5 (stable)	6.5 ± 0.5	12.5 ± 0.5 (stable)
2:3	1.5 ± 0.5 (stable)	7.0 ± 0.5	12.5 ± 0.5 (stable)
3:2	1.5 ± 0.5 (stable)	7.0 ± 0.5	12.5 ± 0.5 (stable)
4:1	1 ± 0.5 (stable)	6.5 ± 0.5	13.0 ± 0.5 (stable)
5:0	1.5 ± 0.5 (stable)	6.5 ± 0.5	12.5 ± 0.5 (stable)

**Table 2 molecules-29-02099-t002:** Acid and alkali resistance of polyurethane emulsion at different mass fractions of UV absorber (before primary chain extension).

Mass Fraction	Acid pH Resistance	Neutral	Alkali pH Resistance
1%	1.5 ± 0.5 (stable)	7.0 ± 0.5	12.5 ± 0.5 (stable)
3%	1 ± 0.5 (stable)	6.5 ± 0.5	12.5 ± 0.5 (stable)
5%	1 ± 0.5 (stable)	7.0 ± 0.5	12.5 ± 0.5 (stable)
7%	1.5 ± 0.5 (stable)	7.0 ± 0.5	12.5 ± 0.5 (stable)
9%	1 ± 0.5 (stable)	7.0 ± 0.5	12.5 ± 0.5 (stable)

**Table 3 molecules-29-02099-t003:** Acid and alkali resistance of polyurethane emulsion at different mass fractions of UV absorber (before water addition).

Mass Fraction	Acid pH Resistance	Neutral	Alkali pH Resistance
1%	1 ± 0.5 (stable)	7.0 ± 0.5	12.5 ± 0.5 (stable)
3%	1.5 ± 0.5 (stable)	6.5 ± 0.5	12.5 ± 0.5 (stable)
5%	1.5 ± 0.5 (stable)	7.0 ± 0.5	12.5 ± 0.5 (stable)
7%	1 ± 0.5 (stable)	7.0 ± 0.5	12.5 ± 0.5 (stable)
9%	1 ± 0.5 (stable)	7.0 ± 0.5	12.5 ± 0.5 (stable)

**Table 4 molecules-29-02099-t004:** Precipitate volume of emulsions at different molar ratios (1,4-butanediol: 2-hydroxyethyl disulfide).

Mole Ratio	0:5	1:4	2:3	3:2	4:1	5:0
precipitation capacity	Lesser	Lesser	Not have	Not have	Not have	Not have

**Table 5 molecules-29-02099-t005:** Investigation plan for the position and concentration of UV absorber addition.

UV Absorber Addition Position	The Addition Ratio of Ultraviolet Absorbents (Based on the Theoretical Solid Content during the Polyurethane Synthesis Process)				
Before the first chain extension	1%	3%	5%	7%	9%
Before emulsification with water	1%	3%	5%	7%	9%

**Table 6 molecules-29-02099-t006:** Investigation plan for the ratio of BDO to 2-hydroxyethyldisulfide addition.

UV Absorber Addition Position	BDO: 2-hydroxyethyldisulfide (Molar Ratio)					
Before the first chain extension	0:5	1:4	2:3	3:2	4:1	5:0

**Table 7 molecules-29-02099-t007:** GB/T 1766-2008 Yellowing Rating Metrics.

Yellowing Rating/Grade	Color Difference Range	Extent of Change
0	E ≤ 1.5	No color change
1	1.5 < E ≤ 3.0	Very slight color change
2	3.0 < E ≤ 6.0	Slight color change
3	6.0 < E ≤ 9.0	Noticeable color change
4	9.0 < E ≤ 12.0	Considerable color change
5	E > 12	Severe color change

## Data Availability

The authors declare that this review is original, has not been published before and is not currently being considered for publication elsewhere. All the authors have checked the manuscript and approved it for submission. My manuscript and associated personal data will be shared with Research Square for the delivery of the author dashboard. Data or code is available to readers and how it can be accessed through the corresponding author.

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
