# Peer review of "Preparation of Yellowing-Resistant Waterborne Polyurethane Modified with Disulfide Bonds"

_molecules, 2024, doi:10.3390/molecules29092099_

Round 1

Reviewer 1 Report

Comments and Suggestions for Authors

The manuscript by G. Li et al. describes the use of a UV absorber to improve the stability of waterborne polyurethanes. Despite the interest of the topic, the manuscript is not suitable for publication, since it lacks of scientific rigour in many descriptions, figures and captions, making the reading too difficult.

Here are some comments:

In the abstract, the nature and the role of the components is not well explained.

“adding 1% to 3% mass fraction” and “1,4-butanediol to 2-hydroxyethyldisulfide 3:2” both lead to a color change of 2? Please clarify

Why the introduction refers to “metallic materials”?

The sentence “without double bonds in the benzene ring” makes no sense. Maybe it is better “without aromatic moieties”?

What does “prone-to-oxidation ether linkages on the α-carbon” means? Does it refer to the homolytic cleavage of the C-H bond in alfa to the ether groups? Please add a reference to clarify.

“Moreover, insufficient surface coordination endows SiO2 nanoparticles with enhanced reactivity to adsorb certain chromophoric species, thus reducing photo-bleaching under UV irradiation.”: Please explain and add a reference.

“Additionally, the UV absorber UV-320 is introduced to synergistically improve the yellowing resistance of polyurethane”: this is a redundant sentence

In the materilals section, what does chemically pure means? Purity numbers should be given when necessary.

Fig 2 is of poor quality (bond angles, in particular, are randomly reported)

Scientific captions should include more details on the meaning of graph and curves.

What does “pre- expansion of one strand“ and “pre-watering” means? This should be clearly stated at the beginning of the Results and discussion part.

In Figure 4, the y scale is missing.

The explanation on the viscosity increase (Fig. 5) is not convincing, especially “the resultant decline in molecular chain length lowers molecular weight, thereby diminishing molecular dimensions and augmenting viscosity”. A reference is needed. Moreover, has the 7% sample a strange behavior due to different sample preparation? Why it was not prepared in the same way? Error bars would have made this graph (and others) more significant.

A reference should be added for “Divergent from anionic and cationic aqueous polyurethanes, nonionic aqueous polyurethanes, devoid of hydrophilic anionic and cationic groups, consequently demonstrate superior acid and alkali resistance capabilities relative to their anionic and cationic aqueous polyurethane counterparts”

Figures 7 and 8 are missing

In figure 9, how was the specific area measured? Details are missing also in the Experimental section

The part “When considered in conjunction with the color difference per Figure 7, this demonstrates enhanced polyurethane resistance to yellowing from the disulfide addition, attributable to the color deviation occurring within the UV-visible range. Aside from the 230 nm wavelength that damages polyurethane, other wavelengths severely degrade the material. The introduction of a disulfide compound enables reversible disulfide bond alterations to absorb higher wavelength visible light at 400 nm as well, improving yellowing resistance.” Is confused. Especially because it refers to a missing Figure 7. Moreover, what does “Aside from the 230 nm wavelength that damages polyurethane, other wavelengths severely degrade the material.” means? . Also “enables reversible disulfide bond alterations to absorb higher wavelength visible light at 400 nm” sounds strange.

In Fig 11, the y scale is missing. Is it arbitrary? Are the spectra normalized?

In figure 18, the S-S bond cleavage is represented only in the scheme but not in the molecular structure

Comments on the Quality of English Language

Some sentences sound strange. I recommend revision by an expert.

Reviewer 2 Report

Comments and Suggestions for Authors

This work presents a comprehensive study on the development of a novel waterborne polyurethane with improved resistance to yellowing. The research methodology is well-structured, and the results contribute valuable insights into the field of polymer science. However, there are a few minor issues that could be addressed to enhance the quality of the manuscript. Below are the specific revision suggestions:

1.     The font format of table 1, table 2 and table 7 is inconsistent with the text and needs to be modified.

2.     In section 1.3.8, the temperature range "300℃ to 700℃" should be changed to "300 ℃ to 700 ℃" with proper spacing. In section 1.3.10, the temperature range "100°C to 150°C" should be changed to "100 °C to 150 °C" with proper spacing.

3.     In section 2.7, the authors should state the corresponding discoloration grade for each color difference result according to the provided table for better clarity.

4.     In the conclusion, the statement "amounting to a discoloration grade of 1" should be corrected to "amounting to a discoloration grade of 2."

5.     Some recent advances can be introduced in the revised backgrounds. I recommend authors to add the following articles in the introduction section which are very useful to enrich their contribution:

Nanodiamond reinforced self-healing and transparent poly(urethane-urea) protective coating for scratch resistance, International Journal of Smart and Nano Materials

Self-healable and reprocessable silicon elastomers based onimine-boroxine bonds for flexible strain sensor, Molecules

Author Response

This work presents a comprehensive study on the development of a novel waterborne polyurethane with improved resistance to yellowing. The research methodology is well-structured, and the results contribute valuable insights into the field of polymer science. However, there are a few minor issues that could be addressed to enhance the quality of the manuscript. Below are the specific revision suggestions:

Reply: We thank the reviewer for carefully reading our manuscript and the support of publication after minor revisions in Molecules. We especially appreciate the insightful questions raised by the reviewer, and welcome the opportunity to address these questions.

  1. The font format of table 1, table 2 and table 7 is inconsistent with the text and needs to be modified.

Reply: Thanks for the kind suggestions of the reviewer. In the revised edition, alterations have been made to the format of the table.

Page 5, Line 1

UV Absorber Addition Position

The addition ratio of ultraviolet absorbents (based on the theoretical solid content during the polyurethane synthesis process)

Before the first chain extension

1%

3%

5%

7%

9%

Before emulsification with water

1%

3%

5%

7%

9%

Table 1. Investigation plan for the position and concentration of UV absorber addition.

Page 5, Line 7

Table 2. Investigation plan for the ratio of BDO to 2-hydroxyethyldisulfide addition.

UV Absorber Addition Position

BDO: 2-hydroxyethyldisulfide (molar ratio)

Before the first chain extension

0:5

1:4

2:3

3:2

4:1

5:0

Page 13, Line10 

Table 7. Precipitate volume of emulsions at different molar ratios (1,4-butanediol: 2-hydroxyethyl disulfide).

mole ratio

0:5

1:4

2:3

3:2

4:1

5:0

precipitation capacity

Lesser

Lesser

Not have

Not have

Not have

Not have

(Seldom: around 1/20 of a 5 mL centrifuge tube; Lesser: around 1/15 of a 5 mL centrifuge tube; More: around 1/10 of a 5 mL centrifuge tube; Enough: around 1/7 of a 5 mL centrifuge tube).

  1. In section 1.3.8, the temperature range "300℃ to 700℃" should be changed to "300 ℃ to 700 ℃" with proper spacing. In section 1.3.10, the temperature range "100°C to 150°C" should be changed to "100 °C to 150 °C" with proper spacing.

Reply: Thanks for the helpful suggestion from the reviewer. In the revised version, revisions have been made to the symbols.

Page 9, Line 2

2.3.8. Thermogravimetric Analysis

The thermal stability of the polyurethane samples was assessed using the TGA4000 thermogravimetric analyzer from PerkinElmer. All samples were thoroughly dried for 24 hours prior to testing. Each test involved weighing approximately 10 mg of polyu-rethane film samples, with an acceptable deviation of ±0.5 mg. The polyurethane film was then placed in a clean vessel and positioned within the instrument for testing. The testing environment parameters were adjusted, with the test gas atmosphere being ni-trogen and the heating rate set at 10 K/min, ramping up from 300 °C to 700 °C.

Page 9, Line 17

2.3.10. Dynamic Mechanical Analysis

Specimens were shaped into thin strips measuring approximately 3 mm in length, 5 mm in width, and 1 mm in thickness. Subsequently, the dynamic thermomechanical analysis (DMA) technique was employed on a TA Instruments Q800 with a tension membrane mode, utilizing a frequency of 1 Hz and a strain amplitude of 1%. The heating rate was set at 5°C/min, spanning a temperature range from -100 °C to 150 °C.

  1. In section 2.7, the authors should state the corresponding discoloration grade for each color difference result according to the provided table for better clarity.

Reply: Thanks for the kind suggestion of the reviewer. The text of section 3.7 has been revised to reflect the following: The sample with a mass fraction of 1% exhibited the best performance, showing only minimal color changes with ΔE=3.2, corresponding to a color change rating of 2. As the ΔE values for samples ranging from 1% to 9% increased, they remained below the blank value. In the case of the pre-water addition UV-320 blending scheme, the sample at 7% demonstrated the smallest total color difference with ΔE=2.6, corresponding to a color change rating of 1, displaying only very slight color variations.

Page 16, Line 1

From Figure 10, it is evident that upon contrasting the pre-crosslinking and pre-water addition samples with the blank group, it is observed that under 300 W ultraviolet lamp exposure for 3 hours and 6 hours, the total color difference ΔE of the blank group is significantly higher than the other five gradients. According to the color change rating range established by GB/T 1766-2008, the blank group undergoes distinct color variation. In the scenario where the UV-320 ultraviolet absorber is introduced before the initial crosslinking, the 1% mass fraction demonstrates the most favorable outcome, with ΔE=3.2, signaling a color change rating of 2, with only a slight chromatic shift. The total color differences from 1% to 9% exhibit an increasing trend but all remain below the blank group. In the scheme where the UV-320 ultraviolet absorber is added before the water inclusion, the sample with a 7% mass fraction displays minimal total color difference, ΔE=2.6, and a color change rating of 1, evidencing only very slight color variation. Compared to the blank group, the color difference within the 1% to 9% range shows minimal variation, indicating a relatively robust resistance to yellowing.

  1. 4. In the conclusion, the statement "amounting to a discoloration grade of 1" should be corrected to "amounting to a discoloration grade of 2."

Reply: Thanks for the kind suggestion of the reviewer. We have made modifications to this section.

Page 20, Line 29

The inclusion of 1% to 3% by mass fraction of the ultraviolet absorbent in the prepoly-mer, in conjunction with the addition of 2-hydroxyethyl disulfide during chain exten-sion, resulted in only slight discoloration, amounting to a discoloration grade of 2, and exhibited substantial resistance to yellowing.

  1. 5. Some recent advances can be introduced in the revised backgrounds. I recommend authors to add the following articles in the introduction section which are very useful to enrich their contribution.

Reply: Thanks for the kind suggestion of the reviewer. We have supplemented the background with additional references.

Page 2, Line 9

However, waterborne polyurethanes are susceptible to yellowing under environmental exposure during production and service. Yellowing impairs product aesthetics, evi-denced by the discoloration of ubiquitous transparent cell phone cases after some du-ration of use[5-11]. In daily life scenarios, white polyurethane shoe materials and some rubber polymer coatings also inevitably undergo yellowing under sunlight irra-diation, severely hindering the manufacturing and use of waterborne polyurethanes in hot and tropical climates. Therefore, enhancing the yellowing resistance of waterborne polyurethanes is of great necessity[10,12-15].

[11] Wang Z, Cao W, Sun C, et al. Nanodiamond reinforced self-healing and transparent poly (urethane–urea) protective coating for scratch resistance[J]. International Journal of Smart and Nano Materials, 2023: 1-20.

[13] Wang P, Wang Z, Liu L, et al. Self-Healable and Reprocessable Silicon Elastomers Based on Imine–Boroxine Bonds for Flexible Strain Sensor[J]. Molecules, 2023, 28(16): 6049.

Round 2

Reviewer 1 Report

Comments and Suggestions for Authors

The authors made a sufficient work to improve the manuscript and can be accepted. I must stress again, however, that scientific captions should be more detailed than those reported in the manuiscript, so to help the reader's understanding (even if details are collected elesewhere in the text). 

Examples:

Figure 4. Raman spectrum of yellowing-resistant polyurethane. (laser wavelenght =?) 

Figure 6. Precipitation (determined as...?) of polyurethane emulsion with different UV-320 content.

Figure 7. Relationship between different mass fractions and specific surface area (determined as...?).

Figure 9. UV-Visible absorption spectrum [collected from (how were them prepared?) samples of] polyurethane with different materials (additives? how much?) .

Moreover,

I guess that in " Then, the sample to be tested was slowly dripped into the container until the measured concentration value on the instrument was in the range of 10 ~ 60 If the concentration data is less than 10..." concentration units are missing.

Comments on the Quality of English Language

Please check the added paragraphs for spelling mistakes (e.g. "introdution" , Figure 2)

Author Response

Responses to Decision Letter on Manuscript “Preparation of Yellowing-Resistant Waterborne Polyurethane Modified with Disulfide Bonds”

April 19, 2024

Journal: Molecules

Dear editor:

Thank you for your letter and for the reviewers’ comments concerning our manuscript entitled “Preparation of Yellowing-Resistant Waterborne Polyurethane Modified with Disulfide Bonds”. We have seriously revised our manuscript according to the reviewers’ suggestions. We annotate a copy of the manuscript to show revisions and track changes for the convenience of the reviewers. All the revisions have been highlighted with a yellow background in the uploaded files named “molecules-2932345-marked”. Subsequently, we will answer each of the reviewers’ comments in earnest. Hope these will make it acceptable for publication. Thank you for your endeavor!

Best Regards,

Dr. Jiale Qu & Prof. Baorong Duan (on behalf of all the authors)

Binzhou Medical University & Yantai University

Response to Reviewer

The authors made a sufficient work to improve the manuscript and can be accepted. I must stress again, however, that scientific captions should be more detailed than those reported in the manuiscript, so to help the reader's understanding (even if details are collected elesewhere in the text).

Reply: We thank the reviewer for carefully reading our manuscript and the support of publication after minor revisions in Molecules. We especially appreciate the insightful questions raised by the reviewer, and welcome the opportunity to address these questions.

Figure 4. Raman spectrum of yellowing-resistant polyurethane. (laser wavelenght =?)

Reply: Thanks for the kind suggestions of the reviewer. In the revised edition, we have opted to re-emphasize the laser wavelength parameter.

Page 5, Line 1

2.3.1. Fourier Transform Infrared (FT-IR) and Raman Spectroscopic Characterization

The 1mm thick polyurethane film was sliced and fixed on the sample holder. The thermosetting polyurethane film underwent infrared spectroscopy scanning in the wavelength range of 400 ~ 4000 cm-1 using the ATR mode on the Thermo Scientific Ni-colet iS20 spectrometer. The thermosetting polyurethane film underwent Raman spec-troscopy analysis using the Senterra confocal Raman microscope (Germany, BRUKER), with a laser light source of 532 nm and a spectral range of 400 ~ 4000 cm-1, to verify the successful conjugation of ultraviolet stabilizers and disulfide bonds.

Page 11, Line 1

Figure 4. Raman spectrum of yellowing-resistant polyurethane (with a laser light source of 532 nm and a spectral range of 400 ~ 4000 cm-1).

Figure 6. Precipitation (determined as...?) of polyurethane emulsion with different UV-320 content.

Reply: Thanks for the helpful suggestion from the reviewer. The annotations regarding precipitation grades present in Table 7, concurrently, the elucidation of precipitation grades has been supplemented into the Materials and Methods section in the revised version.

Page 8, Line 14

2.3.4. Emulsion Stability Evaluation

The stability of emulsions is one of the crucial performance indicators for water-borne polyurethane products, with better emulsion stability correlating to longer stor-age time. In this experiment, approximately 4 mL of uniform polyurethane emulsion was taken in a centrifuge tube and subjected to centrifugation at a speed of 5000 r/min for 8 minutes at room temperature. The centrifuged sample was then observed for any signs of layering or precipitation at the bottom. The precipitate quantities are stratified into 5 distinct grades. (Seldom: around 1/20 of a 5 mL centrifuge tube; Lesser: around 1/15 of a 5 mL centrifuge tube; More: around 1/10 of a 5 mL centrifuge tube; Enough: around 1/7 of a 5 mL centrifuge tube).

Page 13, Line 8

Figure 6. Precipitation of polyurethane emulsion with different UV-320 content. (Seldom: around 1/20 of a 5 mL centrifuge tube; Lesser: around 1/15 of a 5 mL centrifuge tube; More: around 1/10 of a 5 mL centrifuge tube; Enough: around 1/7 of a 5 mL centrifuge tube).

Table 7. Precipitate volume of emulsions at different molar ratios (1,4-butanediol: 2-hydroxyethyl disulfide).

mole ratio

0:5

1:4

2:3

3:2

4:1

5:0

precipitation capacity

Lesser

Lesser

Not have

Not have

Not have

Not have

(Seldom: around 1/20 of a 5 mL centrifuge tube; Lesser: around 1/15 of a 5 mL centrifuge tube; More: around 1/10 of a 5 mL centrifuge tube; Enough: around 1/7 of a 5 mL centrifuge tube).

Figure 7. Relationship between different mass fractions and specific surface area (determined as...?).

Reply: Thanks for the kind suggestion of the reviewer.

Page 14, Line 9

Figure 7. Relationship between different mass fractions and specific surface area. (The data were obtained through measurements taken using the BT-9300H instrument in an environment set at 25 degrees Celsius and 60% humidity.)

Figure 8. Polyurethane emulsion with UV absorber at 1%, 3%, and 7% (from left to right) before primary chain extension.

The particulate dimensions of the polyurethane emulsions were ascertained utilizing a BT-9300H laser particle size analyzer. Concurrently with the acquisition of the polyurethane emulsion globule diameters, the specific surface area of the polyurethane emulsions was obtained. As depicted by Figure 7 and 8, all gradient specific surface areas except the 1% sample exceed the blank, potentially attributable to diminished emulsion particle size from ultraviolet absorbent addition, while the negligible deviation of the 1% sample arises due to the lower UV-320 volume. As UV-320 mass fraction increases, emulsion specific surface area displays an overall growth trend. Specific surface area relates to particle size, with finer particles begetting larger area. Within a defined concentration range, smaller average particle size concentrates the emulsion while larger dimensions dilute it. The test emulsions evidenced declining particle size and rising concentration from 1-9%, with particle dimensions closely tied to emulsion state. With water as carrier, minimal specific surface area renders the emulsion transparent and faintly blue, transitioning to translucent white then opaque white with climbing area.

Figure 9. UV-Visible absorption spectrum [collected from (how were them prepared?) samples of] polyurethane with different materials (additives? how much?).

Reply: Thanks for the kind suggestion of the reviewer. We have made modifications to this section.

Page 15, Line 8

Figure 9. UV-Visible absorption spectrum of polyurethane with different materials. (Pure Polyurethane: Unmodified polyurethane; Addition of disulfides only: Polyurethane modified with a ratio of 3:2 of BDO to 2-hydroxyethyl disulfide; UV absorber only: Polyurethane modified with a 3% content of UV absorber; Addition of disulfide compounds and UV absorbers: Polyurethane modified with a ratio of 3:2 of BDO to 2-hydroxyethyl disulfide along with a 3% content of UV absorber. For specific modification methods, please refer to Section 2.2.2.)

In this experiment, the Cary 300 UV-visible spectrophotometer was used to con-duct the sample testing. Three milliliters of the polyurethane emulsion was taken in a cuvette, with a scanning interval of 5 nm and a wavelength range of 800 ~ 200 nm. As depicted in Figure 9, the blank sample solely displays an absorption peak at 230 nm. Comparatively, the addition of a disulfide introduces a 400 nm peak, while the independent and combined incorporation of an ultraviolet absorbent and disulfide manifest high wavenumber absorption peaks absent in the blank sample. When considered in conjunction with the color difference per Figure 11, this demonstrates enhanced poly-urethane resistance to yellowing from the disulfide addition, attributable to the color deviation occurring within the UV-visible range. Aside from the 230 nm wavelength that damages polyurethane, other wavelengths severely degrade the material. The introduction of a disulfide compound enables reversible disulfide bond alterations to absorb higher wavelength visible light at 400 nm as well, improving yellowing resistance.

Moreover, I guess that in " Then, the sample to be tested was slowly dripped into the container until the measured concentration value on the instrument was in the range of 10 ~ 60 If the concentration data is less than 10..." concentration units are missing.

Reply: Thanks for the helpful suggestion of the reviewer. Regrettably, an oversight in written form has occurred, for which we extend our apologies. The revised version now includes the appropriate units.

Page 8, Line 26

Then, the sample to be tested was slowly dripped into the container until the measured concentration value on the instrument was in the range of 10 ~ 60 μg/ml. If the concentration data is less than 10 μg/ml, it indicates that the sample concentration in the container is too low and more sample should be added.
